# Discovery of Single Independent Latent Variable

**Uri Shaham**
Department of Computer Science
Bar-Ilan University
Ramat Gan, Israel
uri.shaham@biu.ac.il

**Jonathan Svirsky**
Faculty of Engineering
Bar-Ilan University
Ramat Gan, Israel
svirskj@biu.ac.il

**Ori Katz**
Electrical and Computer Engineering
Technion
Haifa, Israel
orikats@campus.technion.ac.il

**Ronen Talmon**
Electrical and Computer Engineering
Technion
Haifa, Israel
ronen@ee.technion.ac.il

## Abstract

Latent variable discovery is a central problem in data analysis with a broad range of applications in applied science. In this work, we consider data given as an invertible mixture of two statistically independent components, and assume that one of the components is observed while the other is hidden. Our goal is to recover the hidden component. For this purpose, we propose an autoencoder equipped with a discriminator. Unlike the standard nonlinear ICA problem, which was shown to be non-identifiable, in the special case of ICA we consider here, we show that our approach can recover the component of interest up to entropy-preserving transformation. We demonstrate the performance of the proposed approach in several tasks, including image synthesis, voice cloning, and fetal ECG extraction.

## 1 Introduction

The recovery of hidden components in data is a long-standing problem in applied science. This problem dates back to the classical PCA [44, 20], yet it has numerous modern manifestations and extensions, e.g., kernel-based methods [47], source separation [11, 3], manifold learning [54, 46, 2, 9], and latent Dirichlet allocation [4], to name but a few. Perhaps the most relevant line of work in the context of this paper is independent component analysis (ICA) [23], which attempts to decompose an observed mixture into statistically independent components.

Here, we consider the following ICA-related recovery problem. Assume that the data is generated as an invertible mixture of two (not necessarily one dimensional) independent components, and that one of the components is observed while the other is hidden. In this setting, our goal is to recover the latent component. At first glance, this problem setting may seem specific and perhaps artificial. However, we posit that it is in fact broad and applies to many real-world problems.

For example, consider thorax and abdominal electrocardiogram (ECG) signals measured during labor for the purpose of determining the fetal heart activity. In analogy to our problem setting, the abdominal signal can be viewed as a mixture of the maternal and fetal heart activities, the maternal signal can be viewed as an accurate proxy of the maternal heart activity alone, and the fetal heart activity is the hidden component of interest we wish to recover. In another example from a different domain, consider a speech signal as a mixture of two independent components: the spoken text and the speaker identity. Arguably, the speaker identity is associated with the pitch and timbre, which are independent of information about the textual content, rhythm and volume. Consequently,

36th Conference on Neural Information Processing Systems (NeurIPS 2022).

recovering a speaker-independent representation of the spoken text facilitates speech synthesis and voice conversion.

In this paper, we present an autoencoder-based approach, augmented with a discriminator, for this recovery problem. First, we theoretically show that this architecture of solution facilitates the recovery of the latent component up to an entropy-preserving transformation. Second, in addition to the recovery of the latent component (the so-called analysis task), it enables us to generate new mixtures corresponding to new instances of the observed independent component (the so-called synthesis task). Experimentally, we show both analysis and synthesis results on several datasets, consisting of simulated and real-world data. In particular, we demonstrate the proposed approach on ECG analysis, image synthesis, and voice cloning tasks.

Our contributions are as follows. (i) We propose an easy-to-train mechanism for extraction of a single latent independent component. (ii) We present a simple proof for the ability of the proposed approach to recover the latent component. (iii) We experimentally demonstrate the applicability of the proposed approach in the contexts of both analysis and synthesis tasks. Specifically, we show applications to real-world data from different fields.

## 2   Related Work

The problem we consider in this work could be viewed as a simplified case of the classical formulation of nonlinear ICA. Several algorithms have been proposed for recovery of the independent components, assuming that (i) the mixing of the components is linear, and (ii) the components (with a possible exception of one) are non-Gaussian; this case was proven to be identifiable [10, 13]. The nonlinear case, however, i.e., when the mixing of the independent components is an arbitrary invertible function, was proven to be non-identifiable in the general case [24].

**Identifiable nonlinear ICA.** Hyvarinen et al. [25] have recently described a general framework for identifiable nonlinear ICA, generalizing several earlier identifiability results for time series, e.g., [21, 22, 51], in which in addition to observing the data $x$, the framework requires an auxiliary observed variable $u$, so that conditioned on $u$, the latent factors are independent (i.e., in contrast to being marginally independent as in a standard ICA setting). The approach we propose in this work falls into this general setting, as we assume that the auxiliary variable $u$ is in fact one of the latent factors, which immediately satisfies the conditional independence requirement. For this special case we provide a simple and intuitive recovery guarantee.

Following works have recently extended the framework of Hyvarinen et al. [25] to generative models [30], unknown intrinsic problem dimension [50], and multiview setting [17]. With respect to iVAE [30], we allow for the recovery of high-dimensional components, whereas in iVAE, only one-dimensional components are recovered. In addition, our work presents several important differences: (i) we formulate our recovery guarantee result in terms of entropy-preserving map rather than statistical identifiability. (ii) It allows for a compact proof of the recovery guarantee. (iii) We present experiments on real-world data and comparisons to leading methods per application domain, whereas in [30], the iVAE approach was mostly demonstrated on simulated data.

**Disentangled representation learning methods.** While the main interest in ICA has originally been for purposes of analysis (i.e., recovery of the independent sources from the observed data), the highly impressive achievements in deep generative modeling in recent years have drawn much interest also to the direction of data synthesis (e.g., images) from independent factors. In the research community, this direction is often termed learning of disentangled representations, i.e., representations in which modification of a single latent coordinate in the representation affects the synthesized data by manipulating a single perceptual factor in the observed data, leaving other factors unchanged. In a similar fashion to the ICA case, the task of learning disentangled representations in the general case was proved to be non-identifiable [39]. Several methods for learning disentangled representations have been recently proposed, most of which are based on a variational autoencoder (VAE, Kingma and Welling [32]) formulation, and decompositions of the VAE objective, for example [19, 31, 8, 7, 33, 6, 14]. GAN-based approaches for disentanglement have been proposed as well [8, 5].

**Domain confusion.** Our proposed approach is based on the ability to learn an encoding in which different conditions (i.e., states of the observed factor) are indistinguishable. Such a principle has been

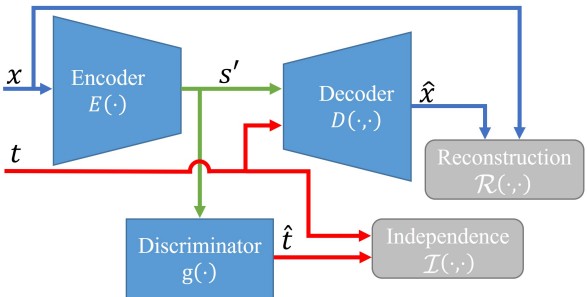

Figure 1: A diagram of the proposed approach. Learned functions are colored in blue, while the objective functions are colored in gray.

in wide use in the machine learning community for domain adaptation tasks [15]. A popular means to achieve this approximate invariance is via discriminator networks, for example, in [35, 42, 48, 43], where the high-level mechanism is similar to the one proposed in this work, although the specifics are different.

We remark that while the algorithm we propose here has been presented before, e.g., in the context of singing voice conversion [43], to the best of our knowledge, it was not discussed in the context of latent independent component recovery, and no identifiability results were shown.

## 3   Problem Formulation

Let $X \in \mathbb{R}^d$ be a random variable. We assume that $X$ is generated via $X = f(S, T)$, where $f$ is an unknown invertible function of two arguments, and $S \in \mathcal{S}$ and $T \in \mathcal{T}$ are two random variables in arbitrary domains, satisfying $S \perp\!\!\!\perp T$. We refer to $S$ as the *unobserved source* that we wish to recover, to $T$ as the *observed condition*, and to $X$ as the *observed input*.

Let $\{(s_i, t_i)\}_{i=1}^n$ be $n$ realizations of $S \times T$, and for all $i = 1, \ldots, n$, let $x_i = f(s_i, t_i)$. Given only input-condition pairs, i.e., $\{(x_i, t_i)\}_{i=1}^n$, we state two goals. First, from analysis standpoint, we aim to recover the realizations $s_1, \ldots s_n$ of the unobserved source $S$. Second, from synthesis standpoint, for any (new) realization $t$ of $T$, we aim to to generate an instance $x = f(s_i, t)$ for any $i = 1, \ldots, n$.

## 4   Autoencoder Model with Discriminator

To achieve the above goals, we propose an autoencoder (AE) with a discriminator. The AE model is denoted by $(E, D)$, where the encoder $E$ maps inputs $x$ to codes $s'$ (analysis), and the decoder $D$ maps code-condition tuples $(s', t)$ to input reconstructions $\hat{x}$ (synthesis), i.e., $x \xmapsto{E} s'$ and $(s', t) \xmapsto{D} \hat{x}$. The discriminator, denoted by $g(\cdot)$, maps codes $s'$ to predicted conditions $\hat{t}$[1].

We use two objective terms in a GAN-like minimax game:

$$\min_{E,D} \max_g \left[ \mathcal{R}\left(x, D(E(x), t)\right) - \lambda \mathcal{I}\left(t, g(E(x))\right) \right]. \tag{1}$$

The first objective, denoted by $\mathcal{R}$, measures the discrepancy between an input $x$ and its reconstruction $\hat{x} = D(E(x), t)$. The second objective, denoted by $\mathcal{I}$, quantifies the independence between the condition $T$ and code $S'$ via a prediction $\hat{t} = g(s')$. In order to maximize $-\mathcal{I}\left(t, g(E(x))\right)$, the discriminator aims at leveraging any information in the code $S' = E(X)$ on the condition $T$ (via a learned function $g$) to obtain an accurate prediction $\hat{t} = g(s')$. In order to minimize both $\mathcal{R}\left(x, D(E(x), t)\right)$ and $-\mathcal{I}\left(t, g(E(x))\right)$, the autoencoder aims at reconstructing the input from the code $s'$ and the condition $t$, while failing the discriminator. We will show formally and empirically, that this results in an equilibrium in which $S'$ does not contain any information on $T$ and contains all the remaining information in $X$. Our approach is illustrated in Figure 1.

---

[1]This formulation of the discriminator does not capture all scenarios, however we use it here for simplicity. In Section 4.2, we describe additional implementations of the discriminator.

## 4.1 Recovery of the Latent Component

Let $S' = E(X)$ be a random variable representing the encoder output, and let $\hat{X} = D(S', T)$ be a random variable representing the decoder output.

Lemma 4.1 establishes that when the autoencoder is trained to perfect reconstruction and the learned code is independent of the condition, the learned code contains the same information as $S$, thereby proving that the latent component of interest can be recovered, up to an entropy-preserving transformation. The lemma is stated assuming $S$ is discrete, and in terms of mutual information $I(\cdot ; \cdot)$ and entropy $H(Y) := -\sum_{y \in \mathcal{Y}} p(y) \log p(y)$, where $Y$ is a random variable with density $p$, taking values in $\mathcal{Y}$. An equivalent result for the case of continuous $S$ can be obtained by replacing the entropy term with the limiting density of discrete points [27] $H(Y) := -\int_{y \in \mathcal{Y}} p(y) \log \frac{p(y)}{m(y)} dy$, where $m(\cdot)$ is the limiting density.

**Lemma 4.1.** *Suppose we train the autoencoder model to zero generalization loss, i.e., $\hat{X} = X$, and impose on the code that $S' \perp\!\!\!\perp T$. Then $I(S; S') = H(S) = H(S')$.*

*Proof.* Since $S \perp\!\!\!\perp T$, $H(S|S') = H(S|S', T) = H(S, T|S', T)$. Since $X = f(S, T)$ and $f$ is invertible, $H(S, T|S', T) = H(X|S', T)$. Since $\hat{X} = X$, and since $\hat{X}$ is a function of $S', T$ we have

$$H(X|S', T) = H(\hat{X}|S', T) = H(\hat{X}, S', T) - h(S', T) = 0.$$

Therefore $H(S|S') = 0$, and $I(S; S') = H(S) - H(S|S') = H(S)$. Finally, since $H(S') \leq H(S)$, we have $H(S) = I(S'; S) \leq H(S') \leq H(S)$, hence $H(S') = H(S)$.

$\square$

Lemma 4.1 has two important consequences. First, it shows that unlike the standard nonlinear ICA problem, the problem we consider here allows for recovery of the latent independent component of interest. More specifically, it proves that when the autoencoder yields perfect reconstruction and condition-independent code, the learned code is a recovery of the random variable $S$, up to entropy-preserving transformation. Second, the lemma prescribes a recipe for the practical solution we present here. Requiring the autoencoder to generate accurate reconstruction $\hat{X}$ of $X$ ensures that no information on $S$ is lost in the encoding process. Independence of $S'$ and $T$ is achieved implicitly; it results from the equilibrium of the GAN-like minimax game (1), as the discriminator can benefit from any mutual information between $S'$ and $T$.

## 4.2 Training Objectives

As described above, to obtain a code $S'$ that is independent of the condition $T$, we utilize a discriminator network, aiming to leverage information on $T$ in $S'$ for prediction, and train the encoder $E$ to fail the discriminator, in a standard adversarial training fashion. Doing so pushes the learned codes $S'$ towards being a condition-free encoding of $X$.

We propose to optimize the following objectives of the discriminator and the autoencoder:

$$\mathcal{L}_{\text{disc}} = \min_g \mathcal{I}\left(t, g(E(x))\right), \tag{2}$$

$$\mathcal{L}_{\text{AE}} = \min_{E, D} \left[\mathcal{R}\left(x, D(E(x), t)\right) - \lambda \mathcal{I}\left(t, g(E(x))\right)\right]. \tag{3}$$

The specific reconstruction and independence objective terms are application-dependent. In our experiments, we make use of the following.

**Reconstruction.** We use standard reconstruction loss functions. In the experiments with images, $\ell_1$ and SSIM loss [56] (and combinations of these) are used. $\ell_1$ loss is also used in the audio experiments, and MSE loss in the experiments with ECG signals.

**Independence.** The discriminator computes a map $\hat{t} = g(s')$, where $s' = E(x)$ is the code obtained from the encoder and $\hat{t} = g(s')$ is the condition predicted by the discriminator. The discriminator is trained to minimize the independence term $\mathcal{I}(\hat{t}, t)$ (thus to leverage mutual information in $S'$ and $T$).

When the condition takes values from a finite symbolic set, we train the discriminator as a classifier that predicts the condition class from the code, and we set the independence term to $\mathcal{I}(\hat{t}, t) =$

Cross Entropy$(\hat{t}, t)$. This is also known as a Domain Confusion term. Using this term, the autoencoder is trained to produce codes that maximize the cross entropy with respect to the true condition, and the equilibrium of the game is when this term equals the cross entropy of a random guess.

When the condition and its prediction take numerical values, i.e., $t, \hat{t} \in \mathbb{R}$, we train the discriminator as a regression model and set the independence term to: $\mathcal{I}(\hat{t}, t) = -\text{Correl}^2(\hat{t}, t)$. We remark that this term also equals the negative $R^2$ term of a simple regression model, regressing $t$ on $\hat{t}$. Using this term, the autoencoder is trained to produce codes for which the squared correlation with the condition is minimized. As $\hat{t}$ is a nonlinear function of $s'$ computed via a flexible model such as a neural net, $\left(\text{Correl}(\hat{t}, t)\right)^2 = 0$ implies that $S'$ and $T$ are approximately statistically independent.

In addition, we also successfully train the discriminator in a contrastive fashion, i.e., to distinguish between "true tuples" $(s', t)$ that correspond to samples $(s, t)$ from the joint distribution of $S$ and $T$ satisfying $s' = E(x)$ and $x = f(s, t)$, and "fake tuples" $(s', t)$, where $s' = E(x)$ but $x = f(s, \tilde{t})$ with $t \neq \tilde{t}$. The contrastive objective is then $\mathcal{I}(s', t) = \text{Cross Entropy}(\hat{l}, l)$, where $l$ is the ground truth true/fake label and $\hat{l} = g(s', t)$ is the predicted true/fake label made by the discriminator. Note that this implementation deviates from the discriminator formulation we used thus far. Here, the discriminator $g(\cdot, \cdot)$ maps tuples of codes and conditions $(s', t)$ to true/fake labels.

We remark that other possible implementations of the independence criterion can be utilized as well, e.g., nonlinear CCA [1, 41] and the Hilbert-Schmidt Independence Criterion (HSIC) [18].

**Optimizers.** Our proposed approach utilizes two optimizers, one for the autoencoder and one for the discriminator. The AE optimizer optimizes $\mathcal{L}_{\text{AE}}$ by tuning the encoder and decoder weights. The discriminator optimizer optimizes $\mathcal{L}_{\text{disc}}$ by tuning the discriminator weights (which determine the function $g$). A common practice in training GANs is to call the two optimizers with different frequencies. We specify the specific choices used in our experiments in Appendix **??**.

**GAN real/ fake discriminator.** Optionally, a GAN-like real / fake discriminator can be added as an additional discriminator in order to encourage generating more realistic inputs. While we have a successful empirical experience with such GAN discriminators (e.g., see Appendix **??**), this is not a core requirement of our proposed approach.

## 5 Experimental Results

In this section, we demonstrate the efficacy of the proposed approach in various settings, by reporting experimental results obtained on different data modalities and condition types, in both analysis and synthesis tasks. We present four applications here and additional two in the appendix.

We begin with a two dimensional analysis demonstration, in which the condition is real-valued. Second, we demonstrate the utility of our approach for image manipulation, where the condition is given as an image. Third, the proposed approach is used for voice cloning, which is primarily a synthesis task with a symbolic condition. Fourth, we apply our approach to an ECG analysis task, using a real-valued heartbeat signal as the condition. Additional experimental results in image synthesis are described Appendix **??** and **??**.

The network architectures and training hyperparameters used in each of the experiments are described Appendix **??**. In addition, codes reproducing some of the results in this manuscript are available at `https://github.com/shaham-lab/disilv`.

### 5.1 2D Analysis Demonstration

In this example, we first generate the latent representation of the data by sampling from two independent uniform random variables. We then generate the observed data via linear mixing. We consider one of the latent components as the condition and train the autoencoder to reconstruct the observed data, while obtaining code which is independent of the condition using the regression objective.We use $\ell_1$ as a reconstruction term. The top row in Figure 2 shows the latent, observed and reconstructed data, as well as the distribution of the condition and the learned code. The bottom row in Figure 2 shows the results of a similar setup, except for the mixing which is now nonlinear. As can be seen, the joint distribution of the learned code and the condition is approximately a tensor product of the marginal distributions, which implies that the latent component is indeed recovered.

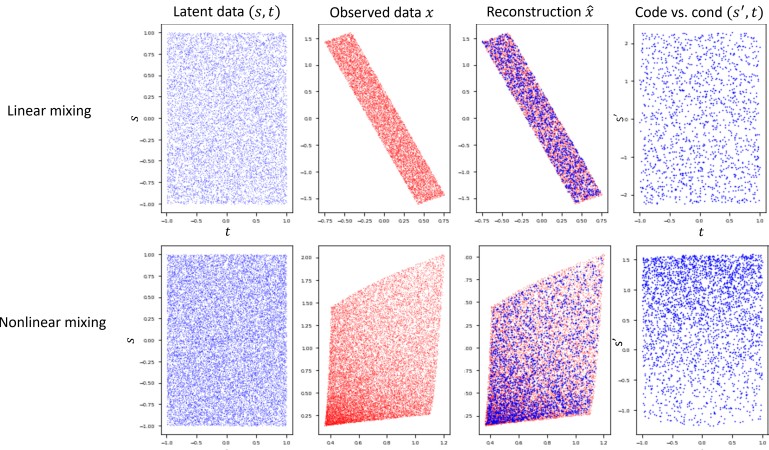

Figure 2: Analysis demonstration. The reconstruction plots show train data in red and reconstructed test data in blue. The learned code $s'$ is a recovery of the latent factor $s$, up to entropy-preserving transformation (e.g., an arbitrary monotonic transformation). The approximate independence of the $S'$ and the condition $T$ can be recognized by noticing that the joint density of the code and the condition is an outer product of the marginal distributions. The $p$-values of $\chi^2$ independence test for the shown results are 0.75 (linear mixing) and 0.83 (nonlinear mixing)

## 5.2 Rotating Figures

In this experiment, we use the setup shown in Figure 3, in which two figures, Bulldog and Bunny, rotate on discs. The rotation speeds are different and are not an integer multiple one of the other. The figures are recorded by two static cameras, where the right camera captures both Bunny and Bulldog, while the left camera captures only Bulldog. The cameras operate simultaneously, so that in each pair of images Bulldog's position with respect to the table is the same. This dataset was curated in [36].

We consider images from the right camera (which contain both figures) as the observed input $x$, and the images from the left camera (which only show Bulldog) as the condition $t$. Note that the input can be considered as generated from two independent sources, namely the rotation angles of Bulldog and Bunny. The goal is to use $x$ and $t$ to recover the rotation angle $s$ of Bunny[2].

Once training is done, we use the autoencoder to generate new images by manipulating Bulldog's rotation angle while preserving Bunny's. This is done by feeding $x$ to the encoder, obtaining an encoding $s'$, sampling an arbitrary condition $\tilde{t}$ and feeding $(s', \tilde{t})$ through the decoder. We use $\ell_1$ loss for reconstruction, and contrastive loss to train the discriminator. Namely, we train the discriminator to distinguish between (image, condition) tuples, which were shot at the same time, and tuples which were not. Figure 4 shows an exemplifying result. As can be seen, the learned model disentangles the rotation angles of Bunny and Bulldog and generates images in which Bunny's rotation angle is preserved while Bulldog's is manipulated.

## 5.3 Voice Cloning

To demonstrate the application of the proposed method to voice conversion, we run experiments on a non-parallel corpus, CSTR VCTK Corpus [55], which consists of 109 English speakers with several accents (e.g., English, American, Scottish, Irish, Indian, etc.). In our experiments, we use a subset of the corpus containing all the utterances for the first 30 speakers (p225- p256, without p235 and p242).

We construct the autoencoder to operate on Mel spectrograms using the speaker id as the condition. The AE architecture was based on Jasper [38] blocks (specific details can be found in Appendix **??**). The decoder uses a learnable lookup table with 64-dimensional embedding for each speaker. For the discriminator, we use the same architecture as in [42]. We use $\ell_1$ loss for reconstruction, and the discriminator is trained using domain confusion loss. Along with the reconstruction loss, in this

---

[2]A related work on this dataset was done in [49], although there the goal was the opposite one, i.e., to recover the common information of the two views, which is the rotation angle of Bulldog.

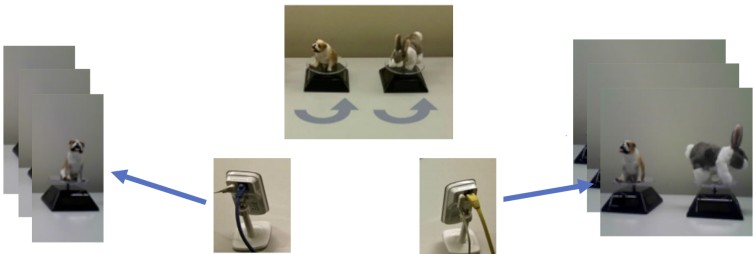

Figure 3: Experiment setup of the rotating figures. Bulldog and Bunny rotate in different speeds. The right camera captures both Bulldog and Bunny, while the left camera captures only Bulldog. Images from the right camera are considered as the input $x$, which is generated from two independent factors – the rotation angles of the figures. Bulldog is considered as the condition $t$. The goal is to recover the rotation angle of Bunny, and to manipulate a given input image $x$ by plugging in a different condition than the one present in the image. The fact that $t$ and $x$ are captured from two different viewpoints prevents modification of the image simply by pasting Bulldog into $x$.

| org image | condition | reconstruction | new condition | synthesis |
| --- | --- | --- | --- | --- |

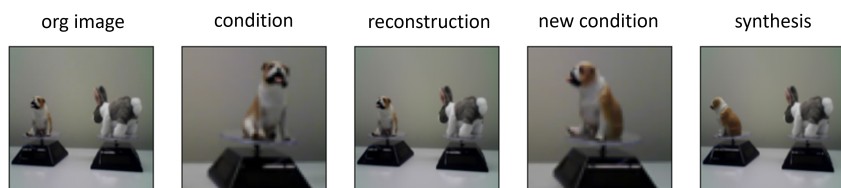

Figure 4: The rotating figures experiment. From left to right: (i) the input image to the encoder $x$, (ii) the condition $t$ corresponding to Bulldog in $x$ as captured from the viewpoint of the left camera, (iii) the reconstruction $\hat{x}$ of $x$, (iv) a new condition plugged into the decoder, and (v) the resulting manipulated image.

experiment we also train a real/fake discriminator, applied to the output of the decoder. To convert the decoder output to waveform, we use a pre-trained melgan [34] vocoder.

Once the autoencoder is trained, we apply it to convert speech from any speaker to any other speaker. Some samples of the converted speech are given at `https://shaham-lab.github.io/disilv/`. To evaluate the similarity of the converted voice and the target voice, we use MCD (Mel Cepstral Distortion) on a subset of the data containing parallel sentences of multiple speakers. Specifically, MCD computes the $\ell_1$ difference between dynamically time warped instance of the converted source voice and a parallel instance of the target voice, which is a common evaluation metric for voice cloning. We remark that the parallel data are used only for evaluation and not for training the model. We use the script provided in [37] to compute the MCD and compare our proposed approach to [12, 45] and references therein, which are all considered to be strong baselines, trained on the VCTK dataset as well. The results, shown in Table 1, demonstrate that our approach outperforms these strong baselines.

## 5.4 Fetal ECG extraction

In this experiment, we demonstrate the applicability of the proposed approach to non-invasive fetal electrocardiogram (fECG) extraction, which facilitates the important task of monitoring the fetal cardiac activity during pregnancy and labor. Following commonly-used non-invasive methods, we consider extraction of the fECG based on two signals: (i) multi-channel abdominal ECG recordings, which consist of a mixture of the desired fECG and the masking maternal electrocardiogram (mECG), and (ii) thorax ECG recordings, which are assumed to contain only the mECG. In analogy to our problem formulation (see Section 3), the desired unobserved source $s$ denotes the fECG, the observed condition $t$ denotes the (thorax) mECG, and the input $x$ denotes the abdominal ECG.

Table 1: Voice cloning results: Mel Cepstral Distortion (MCD) in terms of mean (std). PPG, PPG2 results are taken from [45], VQ AVE and PPG GMM results are taken from [12].

| METHOD | TTS SKINS [45] | GLE [12] | VQ VAE | PPG GMM | PPG | PPG2 | OURS |
|---|---|---|---|---|---|---|---|
| MCD | 8.76 (1.72) | 7.56 | 8.43 | 8.57 | 9.19 (1.50) | 9.18 (1.52) | **6.27** (1.44) |

Table 2: fECG extraction results. In the leftmost column, we present $R_x$, and in the other columns we present $R_{s'}$ achieved by the different methods.

| # OF SUBJECT | INPUT | OURS | ADALINE | ESN | LMS | RLS |
|---|---|---|---|---|---|---|
| TOP 5 | 2.23 (3.23) | **6.86** (1.98) | 6.46 (2.54) | 1.99 (1.08) | 2.60 (1.60) | 1.03 (0.70) |
| TOP 10 | 1.20 (2.41) | **5.43** (2.02) | 4.22 (2.94) | 1.19 (1.10) | 1.56 (1.53) | 0.75 (0.56) |
| TOP 20 | 0.66 (1.75) | **3.53** (2.44) | 2.59 (2.63) | 0.71 (0.91) | 0.89 (1.26) | 0.51 (0.46) |
| ALL | 0.30 (1.17) | **1.84** (2.16) | 1.32 (2.08) | 0.36 (0.68) | 0.40 (0.94) | 0.26 (0.38) |

**Dataset.** We consider the dataset from [52], which is publicly available[3] on PhysioNet [16]. This dataset was recently published and is part of an ongoing effort to establish a benchmark for non-invasive fECG extraction methods. The dataset consists of ECG recordings from 60 subjects. Each recording consists of $n_a = 24$ abdominal ECG channels and $n_t = 3$ thorax ECG channels. In addition, it contains a pulse-wave doppler recording of the fetal heart that serves as a ground-truth. See Appendix **??** for more details.

**Model training.** The input-condition pairs $(x_i, t_i)$ are time-segments of the abdominal ECG recordings ($x_i \in \mathbb{R}^{n_a \times n_T}$) and the thorax ECG recordings ($t_i \in \mathbb{R}^{n_t \times n_T}$), where the length of the time-segments is set to $n_T = 2,000$ (4 seconds). We train a separate model for each subject based on a collection of $n$ input-condition pairs $\{(x_i, t_i)\}_{i=1}^n$ of time-segments.

The encoder is based on a convolutional neural network (CNN), so that the obtained codes $s_i' = E(x_i) \in \mathbb{R}^{n_d \times n_T}$ are time-segments, where the dimension of the code is set to $n_d = 5$. For more details on the architecture, model training, and hyperparameters selection, see Appendix **??**.

We note that the training is performed in an unsupervised manner, i.e., we use the ground-truth doppler signal only for evaluation and not during training.

**Qualitative evaluation.** In Figure 5 we present an example of an input-condition pair $(x_i, t_i)$ and the obtained code $s_i' = E(x_i, t_i)$. We see that the abdominal channels consist of a mixture of the fECG and the mECG, where the fECG is significantly less dominant than the mECG and might even be completely absent from some of the channels. In addition, we see that the thorax channels are affected by the mECG only. Lastly, we see that the obtained code captures the fECG without any noticeable trace of the mECG. In addition, we present the projections of $1,000$ sequentially-sampled inputs $x_i$ (abdominal channels), conditions $t_i$ (thorax channels), and their codes $s_i' = E(x_i)$ on their respective 3 principal components. We color the projected points by the periodicity of the mECG (middle column), computed from the thorax channels, and by the periodicity of the fECG (rightmost column), computed from the ground-truth doppler signals. We see that the PCA of the abdominal and thorax channels are similar, implying that the mECG dominates the mixture. In addition, we see that the color of the PCA of the abdominal and thorax channels according to the mECG (middle column) is similar and smooth, unlike the color by the fECG (rightmost column). In contrast, the PCA of the code is different (bottom row) and only the color by the fECG is smooth (rightmost column), indicating that the code captures the fECG without a significant trace of the mECG, as desired.

**Baselines.** We consider four baselines taken from a recent review [28]. Specifically, we focus on methods that utilize reference thorax channels. The first two baselines are based on adaptive filtering, which is considered to be the traditional approach for fECG extraction: least mean squares (LMS) and recursive least squares (RLS). This approach was first introduced by Widrow et al. [57], and it is still considered to be relevant in recent studies [40, 58, 53]. The third baseline is ADALINE [29] which utilizes neural networks adaptable to the nonlinear time-varying properties of the ECG signal. The fourth baseline is based on an echo state network (ESN) [26].

---

[3]`https://physionet.org/content/ninfea/1.0.0/`

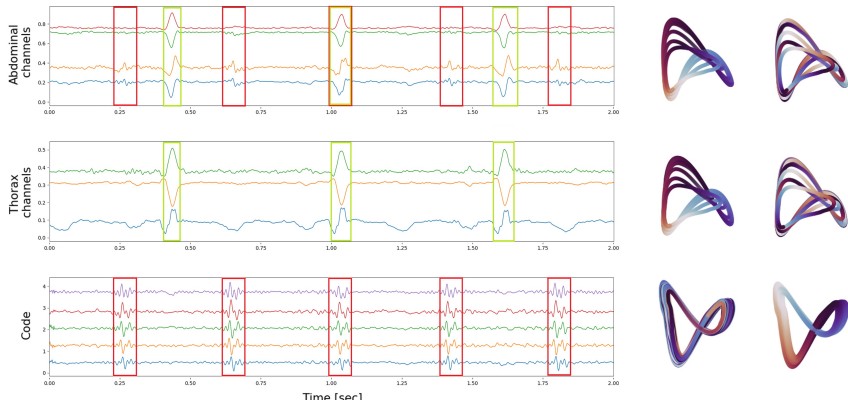

Figure 5: Example of an input-condition pair $(x_i, t_i)$ and the obtained code $s'_i$. The duration of the presented time segment is 2 sec. Leftmost column: the raw channels. Middle column: the PCA embedding of the samples colored by the mECG. Rightmost column: the PCA embedding of the samples colored by the fECG. Top row: the abdominal channels $x_i$ (for brevity only 5 channels are presented). Middle row: the thorax channels $t_i$. Bottom row: the obtained code $s'_i$.

**Quantitative evaluation.** To the best of our knowledge, there is no gold-standard nor definitive evaluation metrics for fECG extraction. Here, based on the ground-truth doppler signal, we quantify the enhancement of the fECG and the suppression of the mECG as follows.

First, we compute the principal component of the input $x_i$. Second, we compute the one-sided autocorrelation of the principal component, and denote it by $A_{x_i}$. Then, we quantify the average presence of the fECG in the inputs $x_i$ by computing: $\bar{A}_x^{(f)} = \frac{1}{n_s} \sum_{i=1}^{n_s} A_{x_i}(\tau_i^{(f)})$, where $\tau_i^{(f)}$ denotes the periods of the fECG obtained from the doppler signals, and $n_s$ denotes the number of time segments in the evaluated recording. Similarly, we compute $\bar{A}_x^{(m)} = \frac{1}{n_s} \sum_{i=1}^{n_s} A_{x_i}(\tau_i^{(m)})$, where $\tau_i^{(m)}$ denotes the periods of the mECG obtained from the thorax signals. Finally, to quantify the relative presences of the signals, we compute the ratio $R_x = \frac{\bar{A}_x^{(f)}}{\bar{A}_x^{(m)}}$. We apply the same procedure to the codes $s'_i$, resulting in $R_{s'}$. When evaluating the baselines, we consider the signals obtained after the mECG cancellation as the counterparts of our code signals.

In Table 2, we present the average ratios in the input, code, and baselines over all the subjects (see Appendix **??** for results per subject). We note that not all the subjects in the dataset include a noticeable fECG in the abdominal recordings. Therefore, we present results over subsets of top $k$ subjects showing highest average ratios $R_x$. We see that our method significantly enhances the fECG with respect to the mixture, and it outperforms the tested baselines.

# 6    Conclusion

In this paper, we present an autoencoder-based approach for single independent component recovery. The considered problem consists of observed data (mixture) generated from two independent components: one observed and the other hidden that needs to be recovered. We theoretically show that this ICA-related recovery problem can be accurately solved, in the sense that the hidden component is recovered up to an entropy-preserving function, by an autoencoder equipped with a discriminator. In addition, we demonstrate the relevance of the problem and the performance of the proposed solution on several tasks, involving image manipulation, voice cloning, and fetal ECG extraction.

Future research will address the limitations of this work. Lemma 4.1 assumes zero generalization loss, i.e., convergence to a global minimum, which is often not achieved in practice. In future work we plan to generalize this statement and assume bounded generalization loss. Another future direction will address noise robustness. The current setting does not consider any noise, either in $t$ or in $x$, or even in $f$. For example, in the presence of noise, perfect reconstruction is undesired, and other losses need to be developed and used.

## Acknowledgments and Disclosure of Funding

We thank the reviewers for their important comments and suggestions. The work of OK and RT was supported by the European Union's Horizon 2020 research and innovation programme under grant agreement No. 802735-ERC-DIFFOP. RT acknowledges the support of the Schmidt Career Advancement Chair in AI.

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
