# Discovery of Single Independent Latent Variable

**Uri Shaham**
Department of Computer Science
Bar-Ilan University
Ramat Gan, Israel
uri.shaham@biu.ac.il

**Jonathan Svirsky**
Faculty of Engineering
Bar-Ilan University
Ramat Gan, Israel
svirskj@biu.ac.il

**Ori Katz**
Electrical and Computer Engineering
Technion
Haifa, Israel
orikats@campus.technion.ac.il

**Ronen Talmon**
Electrical and Computer Engineering
Technion
Haifa, Israel
ronen@ee.technion.ac.il

## A    Colored MNIST experiment

In this experiment, we used a colored version of the MNIST handwritten image dataset, obtained by converting the images to RGB format and coloring each digit with an arbitrary color from {red, green, blue}.

We ran two experiments on this dataset. In the first one we considered the color as the condition. This setup perfectly meets the model assumptions, as each colored image was generated by choosing an arbitrary color at random ($t$) and coloring the original grayscale image ($s$). In the second experiment we set the condition to be the digit label. This corresponds to a data generation process in which handwriting characteristics (e.g., line thickness, orientation) and color are independent of the digit label. While the color was indeed chosen independently of any other factor, independence of the handwriting characteristics and the digit label is debatable, as for example, orientation may depend on the specific digits (e.g., '1' is often written in a tilted fashion, while this is not the case for other digits).

The condition $t$ was incorporated into decoder by modulating the feature maps before each convolutional layer. The discriminator was trained using domain confusion loss. As a reconstruction term we used (pixel-wise) binary cross entropy.

Once the autoencoder was trained, we used it to manipulate the images by plugging to the decoder arbitrary condition and generating new data. Figure 1 shows examples of reconstructions and manipulation for both experiments. In the left panel (showing the results for condition=color) we can see that very high quality reconstruction and conversion were achieved, implying that the learned code did not contain color information, while preserving most of the information of the grayscale image, as desired. The right panel (showing results for condition=digit label) displays similar results, although of somewhat worse conversion quality, as this setting does not fully fit the assumptions taken in this work. Yet, the code clearly captures most dominant characteristics of the handwriting.

## B    Image Domain Conversion

In this experiment we apply the proposed approach to some of the datasets introduced in [**?** ]. Here the condition is the domain (e.g., orange / apple). We use a combination of $\ell_1$ and SSIM loss for reconstruction and domain confusion for independence. In addition to reconstruction loss, we also use a GAN-like real/fake discriminator to slightly improve perceptual loss [**?** ]. Some results are shown in Figure 2. While an interested reader might wonder why oranges are converted to red

36th Conference on Neural Information Processing Systems (NeurIPS 2022).

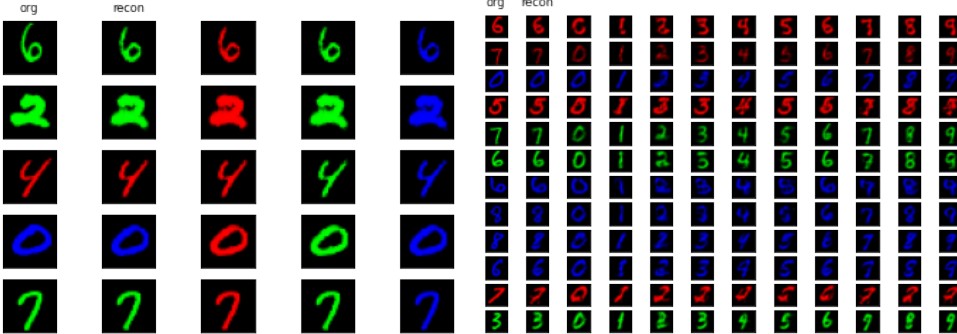

Figure 1: The colored MNIST experiment, using the color (left) and digit label (right) as condition. In each of the plots, the leftmost column show the input $x$ to the encoder, and the next column shows the reconstruction. The remaining columns show conversion to each of the condition classes.

oranges rather than apples, we remark that as much as the condition specifies the type of fruit (orange / apple) throughout this dataset it also specifies its color (orange / red), which, by Ockham's razor, is a somewhat simpler description of the separation between the two domains. Therefore the image manipulation made by the model can be interpreted as a domain conversion.

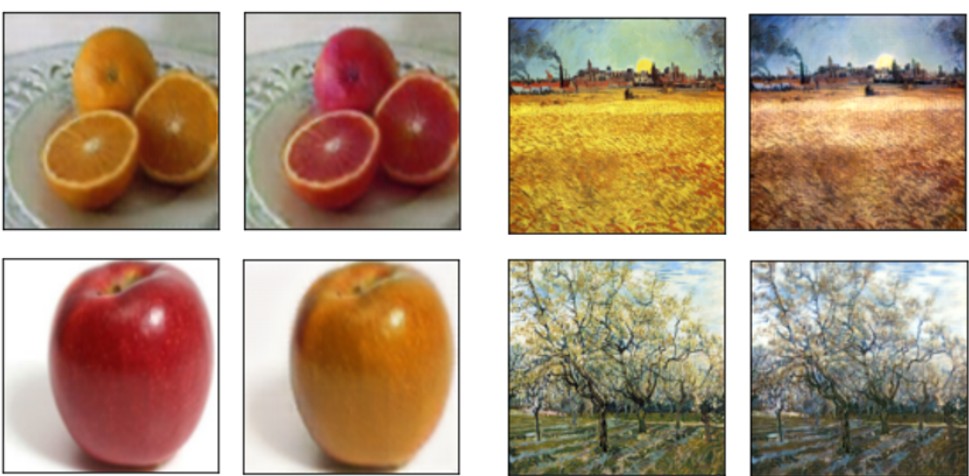

Figure 2: Image Domain Conversion experiment. Left: Conversion results on the oranges2apples dataset. Right: Conversion from Cezanne to photo (up) and Van Goch to photo (down).

## C More Details on the Experiments

### C.1 2D demonstration

In this experiment we used MLP architectures for all networks, where each of the encoder, decoder and discriminator consisted of three hidden layers, each of size 64. Identity and softplus activations were used for the linear and nonlinear mixing experiments, respectively. The discriminator was regularized using r1 loss, computed every eight steps. The model was trained for 100 epochs on a dataset of 15,000 points. To balance between the reconstruction and independence terms we used $\lambda = 0.05$. The autoencoder optimizer was called every 5th step. A Jupyter notebook running this demo is available at https://github.com/shaham-lab/disilv/blob/main/IMAGE/2D_demo.ipynb.

## C.2 MNIST Experiment

In this experiment each of the encoder and decoder consisted of three convolutional layers, of 16, 32 and 64 kernels and ReLU activations. The discriminator had a MLP architecture with 128 units in each layer. The condition was incorporated into the decoder via modulation, utilizing a learnable lookup table of class embeddings of dimension 64. In this experiment we also used an additional discriminator, trained in a GAN fashion to distinguish between real and generated images. During training this discriminator was also trained on swapped data, i.e., codes that were fed to the decoder with a random condition. This discriminator has three convolution layers and was trained with standard non-saturating GAN loss. The system was trained for 50 epochs, with $\lambda = 0.01$ and the real/fake discriminator loss term was added to the $\mathcal{L}_{\text{AE}}$ in (**??**) with coefficient of 0.001. A Jupyter notebook running this demo is available at `https://github.com/shaham-lab/disilv/blob/main/IMAGE/colored_mnist_demo.ipynb`.

## C.3 Rotating Figures

The images $x$ were of size $128 \times 128$ pixels and the conditions were of size $64 \times 64$. In the encoder images were passed through two downsampling blocks and two residual blocks. ResNet encoding models were applied to the condition and the images (separately), before the their feature maps were concatenated and passed through several additional ResNet blocks. In the decoder conditions were downsampled once and passed through two residual blocks, before being concatenated to the codes and fed through two more residual blocks. The decoder and discriminator have a similar architecture. We use $\ell_1$ as reconstruction loss. The system was trained for 120 epochs on a dataset containing 10,000 instances, using $\lambda = 1.05$, and the autoencoder was trained every 5th step.

## C.4 Image Domain Conversion

In this experiment the encoder and decoder's architectures were inspired by the cycleGAN [**?** ] ResNet generator architecture, splitting the generator to encoder and decoder. The decoder was enlarged with modulation layer before each convolutional layer. The class embeddings were of size 512. As in the MNIST experiment, GAN-like real / fake discriminator was used here as well. The system was trained for 200 epochs, on the datasets downloaded from the cyclegan official repository[1]. We used $\lambda_{ind} = \lambda_{rf} = 0.1$ for both discriminators. The autoencoder was trained every 5th step. A Jupyter notebook running this demo is available at `https://github.com/shaham-lab/disilv/blob/main/IMAGE/orange_apples_demo.ipynb`.

## C.5 Voice Conversion

The encoder receives mel-filterbank features calculated from windows of 1024 samples with a 256 samples overlap, and outputs a latent representation of the source speech. The network is constructed from downsampling by factor 2 1D convolution layer with ReLU followed by 30 Jasper [**?** ] blocks, where each sub-block applies a 1D convolution, batch norm, ReLU, and dropout. All sub-blocks in a block have the same number of output channels which we set to 256. The decoder is also a convolutional neural network which receives as input the latent representation produced by encoder and the target speaker id as the condition. The condition was then mapped to a learnable embedding in $\mathbb{R}^{64}$ concatenated to the encoder output by repeating it along the temporal dimension. The concatenated condition is passed through 1D convolution layer with stride 1 followed by a leaky-ReLU activation with a leakiness of 0.2 and 1D transposed convolution with stride 2 for upsampling to the original time dimension. The discriminator was trained using domain confusion loss. The system was trained for 2000 epochs, which took 8 days on a simple GTX 1080 GPU. We used $\lambda = 1$, both optimizers were called every training step.

### C.5.1 Possible negative societal impact

The ability to synthesize realistic audio using voice conversion can be exploited for malicious purposes, e.g., for voice spoofing, fake news, fraud, phishing, and harassment, to name but a few. Therefore, voice conversion should be deployed subject to ethical concerns. Specifically, the converted

---

[1]`https://github.com/junyanz/pytorch-CycleGAN-and-pix2pix/blob/master/datasets/download_cyclegan_dataset.sh`

speech should be presented with an appropriate disclosure indicating that the synthetic speech was generated using a voice conversion technique. In addition, the conversion should be conditioned by obtaining consent from all the affiliated parties (both the source speaker and the target speaker). We acknowledge that this discussion is limited; for a broader discussion we refer the readers to [**?** ].

### C.6 ECG Analysis

#### C.6.1 Dataset

The dataset consists of 60 entries from 39 voluntary pregnant women. Each entry is composed of recordings from 27 ECG channels and a synchronized recording from a trans-abdominal pulse-wave doppler (PWD) of the fetal's heart. The recordings' lengths vary from 7.5 seconds to 119.8 seconds (with average length of 30.6 seconds ± 20.6 seconds). The ECG recordings were sampled by the TMSi Porti7 system with a frequency-sampling rate of 2KHz . The PWD recordings were acquired using the Philips iE33 ultrasound machine. The obtained video was converted into a 1D time series using a processing scheme based on envelope detection. The code for this processing scheme was provided as a Matlab-script by the authors of [**?** ]. For convenience, we uploaded the obtained 1D time series after applying the provided Matlab-script, and it is available at `https://github.com/shaham-lab/disilv/tree/main/ECG/Data`.

#### C.6.2 Pre-proecssing and model implementation

In the following we provide a detailed description of the pre-processing steps and the implementation of the model. For convenience, all the parameters and hyperparameters are summarized in Table 1. The recording of subjects 1-20 were used for hyperparameters selection. These subjects were discarded in the objective evaluation reported in **??**.

**Pre-processing.** The raw ECG recordings were filtered by a median filter with a window length of $n_m = 2,048$ (1 second) to remove the baseline drift. In addition, we apply a notch filter to remove the 50Hz powerline noise and a low-pass filter with a cut-off frequency of $F_c = 125$Hz. Finally, we downsample the signal to frequency-sampling rate of $F_s = 500$Hz. The doppler signal was pre-processed using the script provided by the dataset's owners. No further operations were performed.

**Implementation details.** The encoder module $E(X)$ is implemented using a convolutional neural network (CNN): $\mathbb{R}^{n_a \times n_T} \to \mathbb{R}^{n_d \times n_T}$. This choice of architecture is inspired by the architecture proposed by [**?** ] for the benefit of ECG compression, and it is described in detail in Table 2.

The implementation of the decoder module $D(S', T)$ is based on a deconvolutional neural network (dCNN): $\mathbb{R}^{(n_d + n_t) \times n_T} \to \mathbb{R}^{n_a \times n_T}$. This decoder is applied to the concatenation of the code signal and the thorax signal, where the concatenation is along the first dimension. The exact architecture is described in details in Table 3.

The discriminator is implemented via an additional CNN $g(T) : \mathbb{R}^{n_t \times n_T} \to \mathbb{R}^{n_d \times n_T}$. $g(T)$ shares the same architecture as $E(X)$, except the first convolutional layer which has $n_t$ input channels rather than $n_a$. The independence term is given by $\mathcal{I}(T, S') = \text{Ind}(g(T), S')$, where $\text{Ind}(x, y)$ is a scale-invariant version of the MSE loss function: $\text{Ind}(x, y) = \left|\left| \frac{|x|_e}{||x||_F} - \frac{|y|_e}{||y||_F} \right|\right|_F$, and $|\cdot|_e$ denotes an element-wise operation of absolute value. We remark that other possibilities can be considered as well.

Lastly, the reconstruction module is simply implemented via the standard MSE loss: $\text{Recon}(x, y) = \left|\left| x - y \right|\right|_F$.

#### C.6.3 Training process

We train a model for each subject. The training data is a collection of $n$ input-condition pairs $\{(x_i, t_i)\}_{i=1}^n$, where each input-condition pair $(x_i, t_i)$ is a time-segment that was selected from the ECG recording at a randomly drawn offset and $n$ is a hyperparameter indicating the number of randomly drawn training examples. We use two optimizers that operate in an interleaved (adverserial-like) fashion. Specifically, for each update step of the second optimizer we perform $\beta$ update steps of

Table 1: List of parameters and hyperparameters used in the ECG analysis. Parameters are listed in the upper part of the table, while hyperparameters are listed in the lower part of the table.

| Notation | Description | Value |
|---|---|---|
| $n_a$ | Number of abdominal channels | 24 |
| $n_t$ | Number of thorax channels | 3 |
| $n_m$ | Window length of the median-filter | $2,000$ |
| $F_c$ | Cut-off frequency | $5 \cdot 10^4$ |
| $F_s$ | Frequency-sample | 500 |
| $n_d$ | Dimensioanlity of the code | 5 |
| $n$ | Number condition-pairs for training | $5 \cdot 10^4$ |
| $b$ | Batch size | 32 |
| $lr$ | Learning rate | $10^{-4}$ |
| $\lambda$ | Objective independency factor | 0.01 |
| $\beta$ | Interleaving independency factor | 5 |

Table 2: Layers consisting the encoder $E(X)$ in the ECG analysis.

| No | Layer name | No. of filters $\times$ kernel size | Activation function | Output size |
|---|---|---|---|---|
| 1 | 1D Conv | $8 \times 3$ | Tanh | $2000 \times 8$ |
| 2 | 1D Conv | $8 \times 5$ | Tanh | $2000 \times 8$ |
| 3 | Batch Norm. | - | - | $2000 \times 8$ |
| 4 | 1D Conv | $8 \times 3$ | Tanh | $2000 \times 8$ |
| 5 | Batch Norm. | - | - | $2000 \times 8$ |
| 6 | 1D Conv | $8 \times 11$ | Tanh | $2000 \times 8$ |
| 7 | 1D Conv | $8 \times 13$ | Tanh | $2000 \times 8$ |
| 8 | 1D Conv | $n_d \times 3$ | Tanh | $2000 \times n_d$ |

the first optimizer, where $\beta = 5$ is a hyperparameter. The first optimizer updates $g(T)$ and aims to maximize the dependency between the condition and the code. The second optimizer updates $E(X)$ and $D(S', T)$ and has two objectives – minimizing the reconstruction loss while preventing the first optimizer from succeeding to maximize the dependency loss. Hence, encouraging the optimization process to converge to a "condition-free" code. The proportion between these two objectives is controlled by the hyperparameter $\lambda = 0.01$ which multiplies the second objective term. The losses obtained by the two optimizers are denoted by $\mathcal{L}_{\text{disc}}$ and $\mathcal{L}_{\text{AE}}$ in **??**.

Both optimizers are implemented using the Adam algorithm [? ] with a fixed learning rate of $lr = 10^{-4}$, $\beta = (0.9, 0.999)$ and a bach-size of $b = 32$.

### C.6.4 Qualitative evaluation

Here, we describe in detail the procedure presented in **??** in the paper. First, we produce a code $s'_i$ for each input-condition pair $(x_i, t_i)$. Then, we column-stack each matrix in the set $\{s'_i\}_{i=1}^{1,000}$ and project the obtained set of vectors to a 3D space using principal component analysis (PCA). We repeat the same procedure for $\{x_i\}_{i=1}^{1,000}$. We color the projected points in two manners: according to the fECG signal and according to the mECG signal. The color of the $i$th sample representing the fECG (mECG) signal is computed as follows: $\{\text{mod}(i, \tau^{(f)})\}_{i=1}^{1,000}$ ($\{\text{mod}(i, \tau^{(m)})\}_{i=1}^{1,000}$), where $\tau^{(f)}$ ($\tau^{(m)}$) denotes the period of the fECG (mECG) obtained from the doppler signal (thorax recordings).

### C.6.5 Additional results

The results presented in **??** are averaged over subsets of subjects. In Table 4 we present the results for each subject.

Table 3: Layers consisting the decoder $D(S', T)$ in the ECG analysis. "T.Conv" denotes a transposed convolution layer.

| No | Layer name | No. of filters × kernel size | Activation function | Output size |
|----|------------|------------------------------|---------------------|-------------|
| 1 | 1D T.Conv | $8 \times 3$ | Tanh | $2000 \times 8$ |
| 2 | 1D T.Conv | $8 \times 13$ | Tanh | $2000 \times 8$ |
| 3 | 1D T.Conv | $8 \times 3$ | Tanh | $2000 \times 8$ |
| 4 | 1D T.Conv | $8 \times 5$ | Tanh | $2000 \times 8$ |
| 5 | 1D T.Conv | $n_a \times 3$ | Tanh | $2000 \times n_a$ |

Table 4: fECG extraction results for each subject.

| Subject | Input | Ours | ADALINE | ESN | LMS | RLS |
|---------|-------|------|---------|-----|-----|-----|
| 1 | 0.11 (0.10) | 0.74 (1.63) | 0.51 (0.51) | 0.05 (0.16) | 0.00 (0.00) | 0.20 (0.16) |
| 2 | 0.13 (0.12) | 0.51 (1.90) | 0.17 (0.58) | 0.07 (0.12) | 0.00 (0.00) | -0.02 (0.10) |
| 3 | 0.10 (0.13) | 0.49 (0.90) | 0.56 (0.59) | 0.45 (0.33) | 0.47 (0.33) | 0.04 (0.14) |
| 4 | 0.15 (0.22) | 0.69 (0.68) | 0.10 (0.70) | 0.15 (0.50) | 0.07 (0.15) | 0.08 (0.22) |
| 5 | 7.89 (0.12) | 0.74 (1.04) | -0.01 (0.69) | 0.22 (0.36) | 0.07 (0.72) | 0.04 (0.13) |
| 6 | 0.11 (0.09) | 0.58 (1.14) | 0.39 (0.54) | 0.14 (0.22) | 0.34 (0.45) | 0.10 (0.15) |
| 7 | 0.10 (0.11) | 0.07 (0.99) | 1.04 (0.84) | 0.32 (0.28) | 0.60 (0.41) | 0.09 (0.10) |
| 8 | -0.00 (0.26) | 0.94 (0.69) | 0.26 (0.48) | 0.18 (0.79) | 0.00 (0.00) | 0.17 (0.28) |
| 10 | 0.10 (0.10) | 4.41 (1.48) | 1.46 (2.01) | 0.22 (0.25) | 0.01 (0.02) | 0.03 (0.12) |
| 11 | 0.09 (0.12) | 0.56 (1.73) | 0.68 (0.63) | 0.11 (0.17) | 0.00 (0.00) | 0.15 (0.22) |
| 13 | -0.05 (0.08) | 1.35 (0.50) | 1.08 (1.23) | 0.03 (0.50) | 0.00 (0.01) | 0.58 (0.30) |
| 14 | -0.03 (0.05) | 3.60 (3.18) | 8.82 (4.39) | 1.85 (1.59) | 3.17 (2.50) | 0.90 (0.54) |
| 15 | 0.13 (0.39) | 0.52 (0.90) | 0.36 (0.67) | 0.29 (1.04) | 0.14 (0.82) | 0.28 (0.22) |
| 16 | 0.04 (0.20) | 0.80 (1.57) | 0.20 (0.61) | 0.09 (0.51) | 0.00 (0.00) | -0.03 (0.13) |
| 17 | 0.03 (0.23) | 1.31 (0.45) | 0.29 (0.50) | -0.06 (0.32) | 0.00 (0.00) | 0.58 (0.54) |
| 18 | 0.03 (0.05) | 0.50 (0.82) | 0.58 (1.34) | 0.04 (0.14) | -0.01 (0.02) | 0.12 (0.19) |
| 19 | 0.11 (0.08) | 0.86 (0.94) | 1.03 (0.99) | 0.19 (0.23) | 0.00 (0.00) | 0.22 (0.14) |
| 20 | 0.17 (0.15) | 0.86 (0.94) | 0.24 (0.52) | 0.14 (0.38) | 0.45 (0.54) | 0.28 (0.39) |
| 21 | 0.23 (0.15) | 0.40 (0.45) | 0.28 (0.45) | 0.24 (0.34) | 0.36 (0.45) | 0.44 (0.24) |
| 22 | 1.17 (0.59) | 1.41 (1.10) | 1.00 (1.65) | 1.14 (0.69) | 3.56 (2.09) | 2.28 (2.57) |
| 23 | 0.16 (0.24) | 0.82 (1.31) | 0.17 (1.11) | 0.23 (1.40) | 0.01 (0.03) | 0.03 (0.08) |
| 24 | 0.05 (0.11) | 0.92 (1.05) | 0.19 (0.74) | 0.48 (0.65) | 0.53 (0.35) | 0.28 (0.25) |
| 25 | -0.05 (0.08) | 0.36 (0.38) | 0.81 (0.60) | 0.30 (0.36) | 0.40 (0.53) | -0.05 (0.20) |
| 26 | 0.09 (0.66) | 0.67 (1.08) | 0.56 (0.69) | 0.23 (0.40) | 0.17 (0.34) | 0.11 (0.14) |
| 27 | 0.02 (0.12) | 0.46 (0.71) | 0.34 (0.43) | 0.10 (0.32) | 0.20 (0.40) | 0.09 (0.19) |
| 28 | -0.01 (0.09) | 1.88 (1.80) | 1.55 (1.49) | 0.02 (0.18) | 0.00 (0.00) | 0.15 (0.12) |
| 29 | -0.07 (0.04) | 0.83 (0.90) | 1.09 (0.97) | -0.10 (0.06) | 0.24 (0.04) | 0.04 (0.18) |
| 30 | 0.09 (0.11) | 8.14 (3.21) | 0.95 (0.99) | 0.43 (0.55) | 0.83 (0.97) | 0.34 (0.30) |
| 31 | 0.10 (0.13) | 1.06 (0.60) | 0.33 (0.33) | 0.05 (0.24) | 0.01 (0.00) | 0.01 (0.20) |
| 32 | 0.03 (0.22) | 1.27 (1.04) | 0.25 (0.24) | -0.06 (0.22) | 0.00 (0.00) | -0.05 (0.23) |
| 33 | 0.03 (0.20) | 0.30 (0.35) | 0.35 (0.24) | 0.02 (0.14) | 0.00 (0.00) | -0.09 (0.13) |
| 37 | 0.09 (0.18) | 0.82 (0.70) | 0.02 (0.72) | 0.21 (0.40) | 0.00 (0.00) | 0.15 (0.11) |
| 39 | 0.13 (0.17) | 9.57 (3.24) | 9.16 (1.67) | 0.12 (0.17) | 0.05 (0.02) | 0.48 (0.24) |
| 41 | 0.08 (0.08) | 5.66 (7.76) | 4.28 (3.49) | 0.13 (0.34) | 0.00 (0.00) | 0.32 (0.51) |
| 43 | -0.07 (0.04) | 1.21 (0.80) | 1.61 (1.00) | 0.29 (0.38) | 0.56 (0.37) | 0.35 (0.26) |
| 44 | -0.06 (0.08) | 1.42 (1.03) | 1.09 (1.36) | 0.00 (0.80) | 0.09 (0.09) | 0.27 (0.19) |
| 45 | 1.69 (0.43) | 4.30 (4.73) | 0.56 (1.57) | 0.95 (1.81) | 1.02 (0.68) | 0.77 (0.77) |
| 46 | 0.18 (0.11) | 3.47 (2.77) | 6.45 (5.37) | 3.62 (1.68) | 4.45 (4.05) | 0.17 (0.15) |
| 48 | 0.06 (0.21) | 3.40 (1.68) | 3.12 (2.79) | 2.37 (1.44) | 0.03 (0.02) | 0.24 (0.55) |
| 50 | 0.13 (0.27) | 0.50 (0.79) | 0.52 (0.65) | 0.21 (0.32) | 0.00 (0.00) | 0.01 (0.08) |
| 51 | 0.13 (0.24) | 0.24 (0.48) | 3.59 (2.63) | 0.13 (0.27) | 0.00 (0.00) | 0.65 (0.26) |
| 55 | 0.15 (0.16) | 4.65 (5.32) | 2.15 (1.78) | 0.10 (0.18) | 0.01 (0.00) | 0.04 (0.15) |
| 56 | 0.14 (0.13) | 4.24 (1.60) | 0.77 (0.66) | 0.04 (0.23) | 0.04 (0.11) | 0.08 (0.15) |
| 58 | 0.09 (0.31) | 6.29 (4.99) | -0.01 (0.67) | 0.15 (0.53) | 0.17 (0.62) | 0.45 (0.33) |
| 59 | 0.09 (0.09) | 0.79 (0.88) | 0.34 (0.71) | 0.14 (0.23) | 0.00 (0.00) | 0.35 (0.17) |

### C.6.6  Additional comments

We looked for a dataset that contains: (1) abdominal recordings, (2) chest (thorax) recordings, and (3) ground-truth (GT) that can be used for quantitative evaluation. For this purpose, we reviewed all the datasets from Section 7 in [**?** ]:

- DDB and NIFECGDB: these two datasets do not have GT.
- ADFECGDB: in this dataset, there are no chest recordings.
- PCDB and ADFECGDB: these datasets do not include chest recordings.
- FECGSYNDB: seemingly, this dataset admits all the requirements. However, it is a synthetic dataset, and we were looking for a real-world dataset.

The only real-world datasets that include chest recordings are DDB and NIFECGDB. DDB includes only a single recording of a single subject, and therefore, we focused on NIFECGDB. This dataset was used to objectively evaluate ESN (one of the considered baselines) in [2] using expert annotations.

We contacted the authors of [**?** ] and asked them to share their annotations. Unfortunately, the authors could not share the annotations, but they kindly referred us to use the NIFEADB dataset [**?** ]. This recently published dataset fits our purposes, and therefore, we chose to use it in our experiments.

We note that the GT in NIFEADB is not given as expert annotations of the fetal QRS complexes as in ADFECGDB, PCDB, and the proprietary annotations from NIFECGDB. In NIFEADB, the GT is extracted from the Doppler signal of the fetal heart, making the use of commonly accepted evaluation metrics proposed in [**?** ] impossible for the following reasons:

- Fetal HR measures (listed in the first part of table 5 in [**?** ]: Se,PPV, F1,etc) – these measures assume that the GT includes the locations of the fetal QRS complexes.
- Morphological analysis (listed in the second part of table 5 in [**?** ]: SNR,FQT,TQRS) – these measures assume that the GT includes the fECG signal, which is available only in simulations and in invasive procedures.

Therefore, we used a quantitative evaluation metric that quantifies the enhancement of the fECG and the suppression of the mECG based on the doppler GT.

We remark that the lack of publicly-available reference datasets, which could be used to benchmark different algorithms, was the main motivation for the curation of the NIFEADB (see the abstract in [**?** ]). However, establishing such benchmarks and gold standards is still an ongoing effort, and, to the best of our knowledge, there is no definitive gold standard criterion available to date.