# OpenReview forum: "Discovery of Single Independent Latent Variable"
_NeurIPS.cc/2022/Conference — NeurIPS 2022 Accept_

### Official Review · Reviewer_nS8k · 2022-07-08

**Rating:** 6
**Confidence:** 3
**Soundness:** 4 excellent
**Presentation:** 3 good
**Contribution:** 3 good

**Summary:**

This paper studies the problem of nonlinear independent component analysis in a special case where there are only two components and one is observed. The authors proposed an autoencoder model with a GAN-style additional regularization term which encourages the latent component to be independent to the observed one. The authors proved that the proposed method can recover the latent component up to entropy-preserving transformation. The performance of the proposed method is demonstrated on real world data.


**Questions:**

1. The coefficient for the independence loss lambda takes a different value for each experiment. How is it tuned? Are the results sensitive to the values of lambda?
2. The performance of the proposed method is evaluated in terms of the quality of reconstruction of the same hidden component with a new observed component. This task could also be solved by content-style factorization approaches, although content-style factorization approaches do not force the hidden component to be independent. It would be nice to compare the proposed method with content-style factorization methods.


**Limitations:**

The proof only covers the ideal scenario in which the autoencoder model has zero generalization error, which is impractical.

**Strengths And Weaknesses:**

Strength:
- A new autoencoder-based formulation for discovering one hidden component with nonlinear mixing.
- A theoretical guarantee for the discovery of the hidden component
- The proposed method is evaluated on real world data

Weaknesses:
- The proposed method can only be applied to a restricted nonlinear ICA setting. It can only handle two components, in which one needs to be observed.
- The proof only covers the ideal scenario in which the autoencoder model has zero generalization error, which is impractical.

---

> ### Author Response · Authors · 2022-08-01
> **Response to reviewer nS8k**
>
> $\textbf{Extension to more than two components.}$
> We wish to note that the proposed approach has a natural extension to cases where the observed data is generated as a function of $k$ independent components, of which $k-1$ are observed and the task is to recover the remaining one. This can be done by having a separate discriminator for each one of the $k-1$ observed components.
>
>
> The $\textbf{proof}$ indeed assumes ideal conditions. Yet, this is not uncommon when proving recovery guarantees. For example, the term ``statistical identifiability'' refers to the ability to recover the parameters of a data generation model given an infinite amount of data, which is impractical as well.
>
>
> $\textbf{Choice of lambda.}$ In the toy examples, $\lambda$ was chosen manually to provide nice illustrations.
> In the voice cloning experiment and the ECG experiment, it was tuned manually on the validation subjects.  In our experiments, we have not noticed high sensitivity to the particular choice of $\lambda$. For example, in the ECG experiment, the same $\lambda$was used for all the subjects.
>
> $\textbf{Content-style factorization}$ is indeed an interesting approach to evaluate the generated code. However, please note that it is limited to cases where the two independent factors correspond to content and style. While this may be true in some cases (assuming we are willing to accept some content-style factorization as a ground truth), such factorization does not hold, for example, in our puppets experiment.

---

> > ### Comment · Reviewer_nS8k · 2022-08-09
> > **Re:**
> >
> > Thanks for authors' response.

---

### Official Review · Reviewer_NayA · 2022-07-11

**Rating:** 7
**Confidence:** 3
**Soundness:** 3 good
**Presentation:** 4 excellent
**Contribution:** 2 fair

**Summary:**

This paper studies a special case of non-linear ICA where one of the latent source is known and the goal is to recover the other.
While identifiability results are already known in this setting [27], this paper formulates its guarantees thanks to an entropy-preserving map rather than statistical identifiability and comes with multiple experiments on synthetic and real data (from very different fields).

**Questions:**

- Identifiability: How does the identifiability result compare with [27] ? It seems to me that statistical identifiability is a stronger result than entropy preserving maps. Could your results be derived from [27] by letting the sources by high dimensional ?


- Unclear paragraph:
(l163): "Correl(ˆt, t))2 = 0 implies that S′ and T are approximately statistically independent". In what sense are they approximately statistically independent ? Under which measure of independence ?

(l164-170): Could you explain why training the discriminator in a contrastive fashion leads to minimization of the mutual information between S' and T ? I also believe there is a typo l166 where you wrote $x = (s, \tilde{t})$ I believe it should be $x = f(s, \tilde{t})$. I believe I just need more details to understand this part.

- Other concerns:
 In the legend of Figure 2. Using chi square test this way is wrong, it can only be used to reject independence, not to prove it.

**Limitations:**

Ethical limitations are properly addressed.

**Strengths And Weaknesses:**

Strength:

- The paper is overall very well written. The high level description is very clear, the goals are well described and well motivated.

- The large number of experiments on data from different field shows that the problem is indeed of interest.

- The results are overall convincing

Weaknesses:

- The novelty of the identifiability result is questionable (see the question section)

- l162 -> 170 are a bit unclear (see the question section)

- I tried to open the notebook notebooks/2D_demo.ipynb and it failed with the error
Unreadable Notebook: notebooks/2D_demo.ipynb NotJSONError("Notebook does not appear to be JSON: ''...")

Then, I opened the ECG folder and run Main.py. The code seems to run but with worrying warnings:
padasip/filters/rls.py:183: RuntimeWarning: overflow encountered in matmul
  R1 = self.R @ (x[:, None] * x[None, :]) @ self.R
padasip/filters/rls.py:183: RuntimeWarning: invalid value encountered in matmul
  R1 = self.R @ (x[:, None] * x[None, :]) @ self.R
<class 'ValueError'>
("Input contains NaN, infinity or a value too large for dtype('float64').",)
Input contains NaN, infinity or a value too large for dtype('float64').
In general, the code quality contrasts strongly with the quality of the paper. It does not follow most of the coding guidelines in Python and in particular it comes without any documentation or unit tests. The lack of a clear API prevents easy reproducibility.

In order for others to be able to re-use your work, it would be nice to correct at least the 2D-demo and make it a proper python package with tests, clear documentation, examples (the 2D-demo comes as a perfect example), continuous integration (although I understand this last one is difficult to include in a submission but should definitely be there after publication). It does not take that much time to do and would definitely help other researchers build upon your work.

---

> ### Author Response · Authors · 2022-08-01
> **Response to reviewer NayA**
>
>
> $\textbf{Python Code.}$ First, we apologize for not verifying that the 2D example notebook is working appropriately. We have fixed it and now it does.
>
> Second, we handled the warnings. Please note that these warnings stemmed from the external packages: "Padasip'' and "EchoTorch''  (baselines), and were due to some numerical issues and version incompatibility.
>
> As correctly pointed out by the reviewer, including a proper python package with tests, clear documentation, examples, and continuous integration does not take that much time to do and would definitely help other researchers build upon our work.
>      Upon publication, we will make the code more accessible and user-friendly with a well-documented API and a continuous integration pipeline to encourage other researchers to reuse and further contribute to our work.
>
>
> $\textbf{Identifiability.}$ Please note that statistical identifiability refers to one's ability to recover data generation model parameters in the availability of infinite data. Our recovery result is formulated in terms of information preservation and assumes perfect generalization. These two properties (identifiability and information preservation) focus on different aspects (recovery of model parameters vs. information), yet they both assume infinite training data. We are not sure in what sense the reviewer argues that statistical identifiability is a stronger concept, and would be happy to provide further explanations.
>
> $\textbf{Unclear paragraph (l163).}$ Our statement relies on the fact that random variables $x,y$ are independent if for every two functions $f,g$, $f(x)$ and $g(y)$ are uncorrelated. Since $\hat{t}$ is a function of the code $s'$, computed via a highly expressive mechanism, and trained to leverage any information on $t$, the fact that $\hat{t}$ and $t$ are uncorrelated implies that $s'$ and $t$ are approximately independent.
>     We will make this point clear in the revision.
>
> $\textbf{Training the discriminator in a contrastive fashion.}$ Intuitively, in order to succeed in the contrastive task, the discriminator needs to utilize mutual information between $S'$ and $T$. When such information is available, true and fake pairs can be distinguished. Hence to fail the discriminator, $S'$ and $T$ need to not carry any information on each other, which means that they are independent. The same principle was used by Hyvarinen et al. in [27].
>
> $\textbf{Chi square test.}$ We agree with the reviewer that the Chi square test result implies that the independence assumption is not rejected, which is somewhat different than proving independence. We will add such a note in the revision.

---

> > ### Comment · Reviewer_NayA · 2022-08-08
> > **I am satisfied with the response**
> >
> > Thanks for your response. I am satisfied and updated my grade.

---

### Official Review · Reviewer_gAAc · 2022-07-11

**Rating:** 5
**Confidence:** 3
**Soundness:** 2 fair
**Presentation:** 3 good
**Contribution:** 2 fair

**Summary:**

This paper introduces a novel technique for the extraction of latent variable (or hidden source) form a mixture of independent components. The work is a special case of ICA, extending on the nonlinearity and also offering an identifiable solution. The authors highlight the capabilities on four different applications, including image analysis (synthesis), voice cloning and biomedical data analysis.

**Questions:**

The proposed approach was only evaluated qualitatively and no quantitative evaluation were provided for any of the applications. Could the authors comment why they did not use any quantitative measure (image similarity measure, quality of fetal heart rate monitoring for the fetal ECG application)?
The authors claim that there is no gold-standard for the evaluation metrics for fECG extraction. But that was the purpose of the 2013 PhysioNet/Computing in Cardiology challenge, which proposed metrics for the quality of the fetal heart rate monitoring. The authors could have assessed how the extracted source (fECG) was reliable for estimating the fetal heart rate and if it was in accordance was the fHR estimated from the Doppler signal.
I would also refer the authors to the review Behar, et al. (2016). A practical guide to non-invasive foetal electrocardiogram extraction and analysis. Physiological measurement, 37(5), R1. This review present methods and metrics to be used for the evaluation of fECG extraction techniques.
The interpretation of figures 3, 4, 5 is also really difficult, and readers might struggle to understand the takeaway messages of these figures. For example, the explanation of the qualitative estimation of the fECG problem is given in annex C.6.4, which makes it very hard to understand and interpret the figure 5.
Could the authors comment on how their techniques are coping with noisy measurements, as it seems most of the database used have relatively low level of noise.
Could the authors discuss the impact of the choice of the encoder decoder part of the proposed approach. They suggested the use of a ResNet for image analysis, and simple CNN for fECG analysis, how did they make these decisions.
The caption of the figures should be self-explanatory, (for example figure 1 remind the reader what the different variables stand for (x, t, s’…).
Some acronyms have been introduced without being defined beforehand. For example, line 64, iVAE was introduced without explanations, whereas VAE was introduced and explained on line 82.


**Limitations:**

The potential negative impact of the voice cloning applications are discussed, but this discussion is relegated to the annex, which I believe not to be appropriate.

**Strengths And Weaknesses:**

Strengths: (i) The proposed approach has been tested on several applications, ranging form image analysis, voice cloning, and biomedical data analysis.
(ii) The proposed approach is quite original (to my knowledge), and the use of a discriminator is quite interesting for the extraction of a “disentangled” representation.
Weaknesses: (i) The authors have not evaluated their approach quantitively, and compared their technique with other SOFA techniques. The use of already suggested quantitative metrics is possible for FECG application, and would have been useful.
(ii) The authors did not perform an ablation study, and have not highlighted the added-vale of the discriminator part for analysis purpose. It would have been great to assess a simple VAE for fECG extraction, using the same network architecture to highlight the higher performance introduced by the use of the discriminator.

---

> ### Author Response · Authors · 2022-08-01
> **Response to reviewer gAAc - part 2**
>
> $\textbf{Ablative study.}$
> 1. We thank the reviewer for the suggestion and will add an ablation study in the 2D example. We note that this can be easily performed by setting the parameter $\beta$ in the notebook to zero. We report that the result clearly shows that the discriminator is crucial for obtaining independence.
> 2. Regarding your suggestion to include a comparison to a VAE for the fetal ECG extraction: since the maternal ECG is much more dominant in the abdominal channels compared to the fetal ECG (see the upper plot in Figure 5), we expect that the VAE latent space would mainly encode the maternal ECG. In other words, we expect that without having an explicit restriction for independence between the latent variables and the maternal ECG, the latent variables would mainly parameterize the maternal ECG. This could in fact further demonstrate the necessity of the discriminator in extremely low SNR regimes.
>
> $\textbf{References}$
>
> [1] Behar, Joachim, et al. ``A practical guide to non-invasive foetal electrocardiogram extraction and analysis.'' Physiological measurement 37.5 (2016): R1.
>
> [2] Behar, Joachim, et al. ``An echo state neural network for foetal ECG extraction optimized by random search.'' Proc. Adv. Neural Inf. Process. Syst 36 (2013): 1629-1644.
>
> [3] Pani, D., Sulas, E., Urru, M., Sameni, R., Raffo, L., and Tumbarello, R. (2020). ``NInFEA: Non-Invasive Multimodal Foetal ECG-Doppler Dataset for Antenatal Cardiology Research (version 1.0.0)''. PhysioNet. https://doi.org/10.13026/c4n5-3b04.
>
>
> [4] Yildirim, Ozal, Ru San Tan, and U. Rajendra Acharya. ``An efficient compression of ECG signals using deep convolutional autoencoders.'' Cognitive Systems Research 52 (2018): 198-211.
>
> [5] Zhu, Jun-Yan, et al. ``Unpaired image-to-image translation using cycle-consistent adversarial networks.'' Proceedings of the IEEE international conference on computer vision. 2017.

---

> ### Author Response · Authors · 2022-08-01
> **Response to reviewer gAAc - part 1**
>
> $\textbf{Quantitative evaluation.}$ Please note that we do provide quantitative evaluation for (i) the 2D example (Chi square test result), (ii) the voice cloning experiment (see table 1), and (iii) the fetal ECG extraction.
>
> $\textbf{Fetal ECG extraction.}$ We have put a lot of effort into selecting the dataset and evaluation metric for the fetal ECG extraction. We apologize in advance for the lengthy response, but we wish to convey that the issues raised by the reviewer were carefully considered in our work.
>
> We looked for a dataset that contains: (1) abdominal recordings, (2) chest (thorax) recordings, and (3) ground-truth (GT) that can be used for quantitative evaluation.
>  For this purpose, we reviewed all the datasets from Section 7 in the paper you have mentioned [1] (this reference did not appear in our original submission by mistake and will be added in the revision):
> 1. DDB and NIFECGDB $-$ these two datasets do not have GT.
> 2. ADFECGDB $-$ in this dataset, there are no chest recordings.
> 3. PCDB $-$ this dataset was published as part of the 2013 PhysioNet/Computing in Cardiology Database challenge you mentioned. However, similarly to ADFECGDB, this dataset does not include chest recordings.
> 4. FECGSYNDB $-$ seemingly, this dataset admits all the requirements. However, it is a synthetic dataset, and we were looking for a real-world dataset.
>
> The only real-world datasets that include chest recordings are DDB and NIFECGDB. DDB includes only a single recording of a single subject, and therefore, we focused on NIFECGDB.
> This dataset was used to objectively evaluate ESN (one of the considered baselines) in [2] using expert annotations.
> We contacted the authors of [2] and asked them to share their annotations.
> Unfortunately, the authors could not share the annotations, but they kindly referred us to use the NIFEADB dataset [3]. This recently published dataset fits our purposes, and following their advice, we chose to use it in our experiments.
>
> We note that the GT in NIFEADB is not given as expert annotations of the fetal QRS complexes as in ADFECGDB, PCDB, and the proprietary annotations from NIFECGDB. In NIFEADB, the GT is extracted from the doppler signal of the fetal heart, making the use of commonly accepted evaluation metrics proposed in [1] impossible for the following reasons:
> 1. Fetal HR measures (listed in the first part of table 5 in [1]: Se,PPV, F1,etc) $-$ these measures assume that the GT includes the locations of the fetal QRS complexes.
> 2.  Morphological analysis (listed in the second part of table 5 in [1]: SNR,FQT,TQRS) $-$ these measures assume that the GT  includes the fECG signal, which is available only in simulations and in invasive procedures.
>
> Therefore, we used a quantitative evaluation metric that quantifies the enhancement of the fECG and the suppression of the mECG based on the doppler GT.
>
> Please note that the lack of publicly-available reference datasets, which could be used to benchmark different algorithms, was the main motivation for the curation of the NIFEADB (see the abstract in [3]). However, establishing such benchmarks and gold-standards is still an ongoing effort, and, to the best of our knowledge, there is no definitive gold-standard criterion available to date. In our revision, we will include such an explanation in the appendix with the missing references.
>
>
> $\textbf{Architecture and noise robustness.}$
> The motivation for using ResNet for image analysis and CNN for fECG analysis is based on prior work as follows:
> 1. The choice of the architecture for the fetal ECG extraction was motivated by previous work on ECG compression [4] (see lines 633-635 in the appendix). In [4] the authors propose an auto-encoder for ECG compression, which naturally fits our approach. In addition, in [4] the robustness of the proposed architecture to noise and interference was tested. We rely on their findings and did not include such evaluations in our experimental study.
> 2. The choice of the architecture for the image domain conversion was motivated by the cycleGAN [5] ResNet generator architecture (see lines 581-583 in the appendix).

---

### Official Review · Reviewer_kukj · 2022-07-12

**Rating:** 8
**Confidence:** 3
**Soundness:** 4 excellent
**Presentation:** 4 excellent
**Contribution:** 4 excellent

**Summary:**

This work studies a method to perform non linear ICA where the observed signals $x$ are a non-linear mixing of two independent variables, the sources $s$ and the observed condition $t$: $x = f(s, t)$. The goal of the problem is to recover $s$ and a way to generate new samples $x$ by providing only samples of $x$ and $t$.

To solve the problem, the authors consider an auto-encoder architecture, where the encoder should output the sources $s$ given $x$ as input, the decoder takes the estimated source of the encoder and the condition $t$ as input, and tries to reconstruct the input $x$. A discriminator is also used, to enforce independence between the condition and the estimated sources. Training is performed by solving a min-max optimization problem, where the discriminator tries to minimize independence between the output of the encoder and the condition, while the encoder and decoder are trained to maximize a weighted sum of independence and reconstruction.

The authors demonstrate that if the signals are perfectly recovered by the AE and if the sources $s'$ output by the encoder are independent from T, then $s$ and $s'$ are identical up to an entropy-preserving transform.

The authors then demonstrate the usefulness of the framework on a variety of tasks: synthetic examples, a computer vision disentanglement example, a voice cloning experiment, and a fetal ECG extraction problem.

**Questions:**

- In lemma 4.1, wouldn't it be clearer to write the conclusion as $h(S|S') = 0$?


**Limitations:**

I cannot see potential negative societal impact with this work.

**Strengths And Weaknesses:**

Strengths
-------------
- This paper is extremely well written and pleasant to read. I have learned a lot by reading it.
- The framework proposed is sound and seems novel (even though I am far from an expert on non-linear ICA)
- Theory seems sound
- The experiments are very convincing and clearly demonstrate the wide applicability of the method. They do a great work to show that these ideas go way beyond what was first proposed in [45] for voice conversion.

Weaknesses
-----------------

- The authors could more clearly delineate their work, especially regarding [45], e.g. it should be made clearer that both the architecture in Fig1 and the training objective come from this paper (I find the sentence "We remark that while the algorithm we propose here has been presented before" quite vague)
- The ground truth doppler recording could be displayed in fig.5.


Misc: typo L 218 "lossto"

---

> ### Author Response · Authors · 2022-08-01
> **Response to reviewer kukj**
>
> $\textbf{Relation to [45].}$ As suggested, we will more clearly describe the relationship between our work and [45], and specifically, we will remark that the algorithm and the objective come from [45].
>
> $\textbf{Fig. 5.}$ We will add the doppler signal to Fig 5. Thank you for the nice suggestion.
>
> $\textbf{Lemma 4.1.}$ We formulated the statement in terms of mutual information rather than conditional entropy to more easily associate this result with the discriminator training objective (which uses mutual information to predict $t$). In the revision, we will add a remark that the two formulations are indeed equivalent.

---

> > ### Comment · Reviewer_kukj · 2022-08-06
> > **Response**
> >
> > I thank the authors for their response.

---

### Author Response · Authors · 2022-08-01
**Authors' response to​ all the reviewers**

We thank the reviewers for the time and effort that they invested into the review of our paper, and for their helpful comments and suggestions. Please find our responses below

---

### Meta-Review · Area_Chair_7sxE · 2022-08-27

**Recommendation:** Accept
**Confidence:** Certain

**Metareview:**

Thanks to the authors for this submission, which tackles an interesting and widely appearing problem with a novel approach.  The reviewers agreed that the submission is very well written, motivating the problem and detailing their approach quite clearly.

We also thank the authors for their thorough responses to reviewer questions and comments.  One concern described is the extent to which the fECG experiment realistically and meaningfully evaluates the author’s approach.  Reviewer gAAc expressed some concerns about the quantitative metric used by the authors in the fECG experiment.  Their back and forth revealed the depth of the author’s knowledge of this application area, but did leave some questions unaddressed — namely experiments using simulated ECGs and stress tests on rare observations like premature ventricular contractions, twin pregnancies, or contractions.  Additionally, reviewer gAAc asks a pointed question — how does the heart rate estimate from the extracted fECG and the Doppler signal directly compare?  If I’m not mistaken, this comparison is feasible with the dataset analyzed, and could even be prepared in a subsequent draft.

Despite these open questions, this work does look at a breadth of applications in their experiments, strengthening the submission greatly.  While I recommend accept, I strongly urge the authors to address these last points brought up by reviewer gAAc.

**Award:**

No

---

### Decision · Program_Chairs · 2022-09-14

Accept